# Welfare Assessment in Pigs Using the Salivary Proteome

**DOI:** 10.3390/ani14111703

**Published:** 2024-06-05

**Authors:** Sara Prims, Chris Van Ginneken, Xaveer Van Ostade, Christophe Casteleyn

**Affiliations:** 1Comparative Perinatal Development, Department of Veterinary Sciences, Faculty of Pharmaceutical, Biomedical and Veterinary Sciences, University of Antwerp, Universiteitsplein 1, 2610 Wilrijk, Belgium; sara.prims@uantwerpen.be (S.P.); chris.vanginneken@uantwerpen.be (C.V.G.); 2Laboratory of Protein Science, Proteomics and Epigenetic Signaling (PPES), Department of Biomedical Sciences, Faculty of Pharmaceutical, Biomedical and Veterinary Sciences, University of Antwerp, Universiteitsplein 1, 2610 Wilrijk, Belgium; xaveer.vanostade@uantwerpen.be; 3Department of Morphology, Medical Imaging, Orthopedics, Physiotherapy and Nutrition, Faculty of Veterinary Medicine, Ghent University, Salisburylaan 133, 9820 Merelbeke, Belgium

**Keywords:** pig, piglet, saliva, proteome, proteomics, biomarkers, stress, welfare, health status

## Abstract

**Simple Summary:**

Piglets raised at a farm for meat production experience stress that affects their welfare. Recently, it has been demonstrated that several salivary proteins are up- or down-regulated in stressful situations. These could, therefore, be used as biomarkers for stress. This review revisits the definition of stress and defines the protein composition of porcine saliva to finally propose a panel of six salivary stress biomarkers.

**Abstract:**

Identifying the potential presence of stress at the pig farm is fundamental since it affects pig welfare. As a result, a reliable and straightforward tool to monitor stress could record the welfare status of the animals. Although numerous methods to assess the welfare of pigs have been developed in the past, no gold standard has been established yet. Recently, the value of saliva as a tool to identify chronic stress in piglets was explored, as it can be collected fast and non-invasively. Since the protein composition, i.e., the proteome of porcine saliva, responds to stress, the affected proteins could be used as salivary stress biomarkers. The present review first defines stress and its relationship with welfare. Next, the porcine gland-specific salivary proteome is characterized. Finally, six potential salivary biomarkers for stress are proposed, i.e., odorant-binding protein, vomeromodulin-like protein, chitinase, lipocalin-1, long palate lung and nasal epithelium protein, and alpha-2-HS-glycoprotein.

## 1. Introduction

Pigs in farm settings are commonly exposed to several stressors, such as regrouping [1], feed changes [2], and road transport [3]. When a stressor exceeds a certain threshold in duration and/or magnitude, the body’s homeostasis is disturbed. The equilibrium can be re-established by physiological and behavioral adaptive responses and/or by removing the stressor. However, failure to generate sufficient adaptive responses leads to chronic stress, compromising animal welfare [4]. Identifying stress at the farm is a prerequisite for the evaluation of various management strategies, aiming to increase animal welfare. Therefore, a fast, straightforward, and reliable tool to monitor stress is required. Although numerous methods to assess welfare in pigs are currently available, their limitations have resulted in the lack of a conclusive set of parameters or indicators of animal welfare [4].

Recent studies have explored saliva as a potential source of biomarkers to assess stress in pigs [5]. The present review in this Special Issue entitled Saliva and Blood Markers in Animal Welfare and Health Monitoring focuses on the welfare assessment of pigs using the salivary proteome, i.e., the protein composition of saliva. It is hypothesized that changes in the salivary proteome occur when the welfare status of pigs is altered.

The definition of stress and its relationship with welfare will be scrutinized first. Subsequently, the porcine salivary proteome is described. Since proteomics technologies are best suited for salivary proteome analysis, a brief introduction to proteomics is included. Potential salivary biomarkers for stress in pigs will be presented and the value of salivary stress markers in welfare assessment will be discussed.

## 2. The Definition of Stress

### 2.1. Stress Responses

An individual will experience stress when its homeostasis is threatened. Consequently, stress can be seen as a physical or psychosocial (or metabolic or immunological) force threatening the body’s homeostasis. The subject will attempt to re-establish this homeostasis through physiological and behavioral adaptive responses. The specific responses to the threats will, to some degree, depend on the perceived threat’s character [4]. In other words, no universal stress response exists. Different types of stressors will result in different stress responses. Since the individual’s perceptions of the stressors and the ability to cope with these vary between individuals, the interpretation of stress responses is challenging [6].

A stressor activates the sympathetic adrenomedullary (SAM) axis, leading to the secretion of catecholamines by the adrenal glands. These catecholamines, epinephrine and norepinephrine, are secreted into the bloodstream and induce a rapid physiological adaptation to increase alertness, vigilance, and focused attention to aid the subject in making a strategic decision at the start of a stressful challenge. The physiological changes entail cardiovascular changes and metabolic actions leading to increased blood glucose through glycogenolysis and gluconeogenesis, lipolysis, increased oxygen consumption, and thermogenesis. After the stressor has been dealt with, the physiological and adaptive responses are counteracted by activating the parasympathetic nervous system [7,8].

Simultaneously, the stressor activates a slower response through modulation of the hypothalamic-pituitary-adrenal (HPA) axis. Upon stimulation, neurons in the paraventricular nucleus of the hypothalamus release the corticotropin-releasing hormone (CRH) and other secretagogues such as arginine-vasopressin. The former hormone is transported through the hypophyseal portal circulation to the anterior pituitary gland (adenohypophysis). Here, it initiates the cleavage of pro-opiomelanocortin into adrenocorticotropin (ACTH), β-endorphin, and other peptides, and their subsequent release from the anterior pituitary gland into the bloodstream. The ACTH signal is carried through the peripheral circulation to the adrenal glands, triggering the release of glucocorticoids and adrenal androgens from the adrenal cortex [7,9]. Glucocorticoid receptors are present in almost all tissues of the body. Consequently, cortisol can affect nearly every organ system. Glucocorticoids can have catabolic, lipogenic, immunosuppressive, and anti-reproductive effects and may influence the cardiovascular system and behavior [10]. Circulating glucocorticoids suppress the secretion of CRH from the hypothalamus and directly inhibit ACTH secretion from the pituitary gland. Additional feedback loops include the inhibitory effects of ACTH, β-endorphin, and CRH itself on the hypothalamic CRH neurons. This negative feedback mechanism limits the duration of the body’s exposure to the effects of these glucocorticoids [11].

Activating the SAM and HPA axes in response to a stressor results in the release of various hormones, neuropeptides, and neurotransmitters. Bidirectional communication between the nervous, endocrine, and immune systems through the shared use of ligands and receptors leads to a broad spectrum of biological and behavioral consequences.

### 2.2. Stress and Animal Welfare

Stress can be acute or chronic, referring to the duration that can be short, lasting minutes to various days, or long (lasting weeks, months, or even years), respectively [4]. An acute stress response helps the individual respond and adapt to an acute stressor. It is essential for an individual’s survival and is not harmful. This adaptation mechanism is, however, useless in chronic stress that can result from a single, highly traumatic, acute stressful event or could be induced by the longer duration or higher magnitude of repeated single or multiple stressors [12].

A chronically stressed animal fails to restore its homeostasis and is characterized by elevated glucocorticoid levels. Indeed, chronic stress can lead to hyperreactivity of the adrenal cortex. It can even result in an exaggerated ACTH response to new acute stressors [13]. Since high levels of glucocorticoids are noxious when they persist over a longer period, chronic stress will eventually affect the body in many short-term and long-term ways [14].

Chronic stress can affect the immune system [15], reduce zootechnical performance [16], and disturb breeding capacity [17,18]. Hence, stress and welfare are inevitably connected. Reverse reasoning dictates that a healthy and well-performing animal is faring well. Such an animal enjoys its five freedoms, i.e., freedom from thirst, hunger, and malnutrition, freedom from thermal and physical discomfort, freedom from pain, injury and disease, freedom from fear, distress, and chronic stress, and freedom to express normal behavior [19]. However, in this reasoning on welfare, the animal’s feelings are not recognized. Therefore, Mellor introduced the concept of the five domains of animal welfare [20]. This holds that the animal receives good nutrition, is in thermal and physical comfort, has a good health and physical condition, is free to express normal behavior, and is in a good affective state that promotes the expression of positive emotions and avoids negative emotions such as fear, anxiety, frustration, boredom, or pain). It is obvious that welfare is a complex issue and welfare scientists are challenged to develop methods to score the welfare status of animals.

## 3. Saliva as a Biological Matrix to Assess Welfare

The Welfare Quality protocol rolled out in 2009 to assess welfare at the farm mainly focuses on causal and some consequence indicators such as mortality and weight gain, behavior assessment, and identifying stereotypies and (signs of) wounds/lesions or disease, like coughing and sneezing. Some biological response indicators, for instance, breathing patterns, are included in the protocol, too. However, no other physiological parameters are assessed. Therefore, the existing protocol could greatly benefit from adding physiological assessment(s).

With regards to biological matrices, blood and saliva are the most versatile samples and could provide the largest amount of information on the presence of stress, and since stress and animal welfare are inevitably connected, on the welfare status of animals. Although factors like sex, breed, season, etc. also have implications for the reference values of blood biomarkers, the introduction of variation due to sampling and contamination is negligible for this matrix. However, the invasive nature of blood sampling remains its most decisive disadvantage [4].

In contrast, saliva, urine, feces, and hair collection can be performed non-invasively. Since the latter three biological matrices accumulate biomarkers over a certain period, they are less sensitive than blood and saliva to minor variations such as circadian rhythms [21,22,23,24,25,26]. Hair is the preferred matrix of these three biological matrices since it is less prone to microbial degradation and more straightforward to sample [27,28,29]. However, the major drawback of this matrix is that the variety of analytes that can be measured in hair remains limited. That is where saliva comes into the picture.

## 4. The Salivary Proteome

### 4.1. Proteome Analysis

Proteomic approaches to analyze saliva offer multiple advantages. A major benefit is that both targeted and untargeted applications of proteomics exist. In contrast to targeted proteomic approaches, untargeted analysis of the salivary proteome facilitates hypothesis-free testing. Furthermore, the entire proteome can be studied instantaneously, and differences in salivary protein abundance between treatment groups are highlighted. Using proteomics, up- or downregulation of proteins that are not purposely monitored can also be identified. On the other hand, targeted approaches are also suited to analyze the (relative) abundance of specifically scrutinized proteins. Another plus of proteomics is that it does not rely on antibodies’ availability. Although the use of pigs as research animals is gaining interest, the accessibility of antibodies or antibody-based assays that are specific for porcine proteins is limited compared to those that are developed for use with human antigens or antigens derived from typical animal model species like mice. Notwithstanding the high degree of homology between the amino acid sequences of porcine and human proteins, the cross-reactivity of a human antibody against a porcine antigen is not evident. Therefore, combining untargeted and targeted proteomic approaches to study the porcine salivary proteome is highly esteemed.

Porcine saliva contains a complex mixture of proteins. To identify as many proteins as possible, they must be separated and fractionated first. This can be performed using two-dimensional polyacrylamide gel electrophoresis (2D-PAGE) or multidimensional liquid chromatography (LC). After the complex sample is separated and fragmented, the resulting less complex fractions are further analyzed, and peptides are characterized using mass spectrometry (MS). This technique first ionizes the analytes using an ion source. Next, a mass analyzer sorts the ions according to their masses and charges (mass/charge or *m*/*z* ratio). Finally, a detector measures the abundance of each detectable ion. A more complex form of MS is tandem MS or MS/MS. With this technique, the analyte undergoes two MS rounds with a fragmentation step in between [30,31,32].

Several quantification methods using peptide digests can be combined with LC-MS/MS. Quantification can be accomplished with a global approach to determine the abundance of all peptides in the sample, for example, using isobaric tags (iTRAQ: isobaric tags for relative and absolute quantification) [33]. Alternatively, the abundance of only specific, targeted peptides in samples can be assessed using parallel reaction monitoring (PRM) techniques [34].

Before proteins can be identified, their composing peptides should be identified first from MS and MS/MS spectra. The most applied method is database searching, in which software matches the observed spectra to theoretical or previously generated spectra from known peptides. The accuracy of a possible peptide match is usually calculated and indicates how trustworthy this identification is. Only highly reliable peptide identifications are used for the following step, i.e., protein identification, and it is strongly advised to use several peptides to identify a protein. Different kinds of software can be used to identify the peptide sequences and to match these to protein sequences in generated protein databases.

Only the proteome of the species under investigation can be used to limit the number of possible matches. However, a correct match can only be made if the correct genome sequences that encode for the protein repertoire are in the search database. Consequently, an underestimation of the number of identifiable peptides can result from the incomplete presence of the organism’s genome sequence in the database [32]. Determining the false discovery rate (FDR) is an additional global confidence assessment. To reach this goal, the generated spectra are searched against a target database and against a “decoy” database. The latter is constructed by reversing or shuffling the protein sequences from the target database, leading to peptides with non-existing amino acid sequences. The number of false identifications estimates the reliability of the obtained identifications using the target database. For example, an FDR of 5% means that 5% of the obtained peptide identifications are probably false.

### 4.2. Potential Salivary Biomarkers for Stress in Pigs

In recent years, a number of potential biomarkers for physical and psychosocial stress have been detected in porcine saliva. The involved studies focused on one, or a few, targeted compounds, which were, however, not exclusively proteins. These studies have been reviewed by Prims et al. [35] and the suggested salivary biomarkers for stress in pigs are listed in Table 1.

The above-cited study demonstrated that chronic exposure of piglets to multiple stressors alters the salivary proteome [35]. Shotgun analysis identified 392 proteins in the saliva of 28-day-old piglets. The relative abundance of 20 proteins was affected by three weeks of exposure to multiple stressors. Further analysis of eight proteins confirmed that six are potential biomarkers for chronic stress. To this purpose, saliva samples taken one week and three weeks after the initiation of the chronic exposure to the stressors were analyzed to verify their profiles over time. The targeted PRM analysis confirmed that alpha-2-HS-glycoprotein was upregulated in the stressed group after one and three weeks, while odorant-binding protein, chitinase, long palate lung and nasal epithelium protein 5, lipocalin-1, and vomeromodulin-like protein were present in lower concentrations in the saliva of the stressed pigs, albeit only after three weeks and not yet after one week. These six proteins are listed in the upper rows of Table 1 and are presented in bold font.

Since the answer to whether chronic stress alters the salivary proteome of pigs is positive, the value of the salivary proteome as a tool to monitor stress and, hence, evaluate the welfare status of pigs should be explored. However, since numerous factors can introduce variations in specific protein concentrations, the reliability and comparability of these potential biomarkers can be affected. Assessing how these factors can affect the salivary concentrations of the discovered candidate biomarkers for physical and psychosocial stress is critical.

### 4.3. Factors That Could Introduce Variation in Salivary Biomarker Concentration

The factors that can introduce variation in the concentrations of the salivary biomarkers for physical and psychosocial stress that are listed in Table 1 are discussed below. They are additionally presented in that table.

#### 4.3.1. Saliva Analysis Technique

Salivary biomarker concentrations can be analyzed using different techniques. The most frequently used methods include antibody-based techniques like ELISA (enzyme-linked immunosorbent assay), TR-IFMA (Time-resolved immunofluorometric assay), and alphaLISA^®^ (Perkin Elmer, Malines, Belgium) [36,37,38], enzymatic assays for enzymes like α-amylase and adenosine deaminase [39], and proteomic approaches like HPLC, gel-based or liquid-based separation followed by MS/MS [40]. Although all these techniques have the same purpose, namely determining the quantity of a molecule in a biological sample, some variation in outcome between the different techniques is possible. For example, when cortisol was measured in the same sample with a competitive chemiluminescence enzyme immunoassay and with an indirect competitive alphaLISA^®^ assay, validated for porcine saliva, the cortisol concentrations were 1.5 times higher when determined with the alphaLISA^®^ assay. Additionally, this latter technique has proven to be more sensitive to contamination with fecal matter than the chemiluminescence assay, resulting in the detection of artificially high cortisol values [41].

Similar observations have been made during the analysis of α-amylase in human saliva. The concentrations of this protein were determined either enzymatically or directly with a non-competitive indirect sandwich assay. Not only were the concentrations consistently lower when the enzymatic activity was measured, they also displayed higher inter-individual variability. The correlation between both assays varied depending on the applied acute stressor. It was assumed that specific isoforms of the enzyme were detected with the direct-protein assay, which could not be discovered with the enzymatic assay. Since the difference between the control and the stressed group was more significant when the concentrations were measured with the enzymatic assay, this assay was suggested to be more sensitive for detecting changes induced by acute stress [42].

Although the above-mentioned techniques determine the concentrations in g/mL or units/mL, most MS techniques report the total amount of proteins in concentrations or proportions. Therefore, corrections for the total protein concentration in the sample should be made to enable comparison. More data regarding this normalization can be found in Section 4.3.2.

The lack of a well-characterized porcine protein database hampers the identification of salivary proteins in pigs using MS techniques. When protein identification was performed in 2018 by Prims et al., only 50,045 (reviewed + unreviewed) entries were found in the *Sus scrofa* database accessible through UniProt [43]. Two years later, 120,806 (reviewed + unreviewed) entries were already present [27]. At the end of 2023, a total of 398,092 entries were available at UniProt. However, from these nearly 400,000 entries, only 3,590 proteins are reviewed [35]. These reviewed proteins are manually annotated, combing experimental results, computed features, and scientific conclusions. The unreviewed proteins are based on genome projects and are computational records with automatic annotation. Fortunately, information on the latter, unreviewed proteins, is frequently updated. The incomplete database particularly impedes PRM analysis, a quantification technique that relies on quantifying protein-specific peptides [34]. When unique peptides for the target protein are searched in an incomplete and not fully annotated database, it is advisable to analyze at least three, but preferably more, protein-specific peptides and compare their profiles. If the abundance profile of a protein-specific peptide deviates from that of the other protein-specific peptides, then this peptide probably is not unique to the target protein.

#### 4.3.2. Result Normalization

Normalization is often performed when varying factors could influence the sample. An often-used method is normalizing with a housekeeping protein. Although amylase, mucins, albumin, or IgA have also been suggested, the abundance of these proteins, except for mucins, is altered by acute stress [27,36,44]. Apomucin or sulfhydryl oxidase could also be useful for normalization or identifying a large difference in the background proteome since their variations are limited in stressed piglets [35]. Unfortunately, chewing affects the apomucin concentrations. This can be deduced from the fact that this protein is more abundant in mandibular and sublingual secretions than in parotid saliva [43].

Another often-used method is normalizing for protein secretion rate [42]. This implies that the protein output per unit of time must be measured. Unfortunately, this factor cannot be determined in pigs using sponges to collect saliva since they have a ceiling/saturation limit [5,45].

Furthermore, the total protein concentration has been suggested for normalization. This method is also limited since stress increases the total salivary protein concentration [45]. In addition, chewing may interfere with this factor [35]. Finally, contamination of feed, dirt, and fecal material may also affect this parameter [41].

An alternative method that could reduce the variation arising from a difference in sample processing, diverse salivary flow rates, or sample contamination (e.g., with blood, feed, or dirt) is combining the data instead of considering the proteins separately. This is interesting when control animals are compared with potentially stressed animals in an experimental setting. By dividing the abundance of the upregulated protein by the average of the abundances of all the proteins that had a significant downregulation per potentially stressed animal, one value or ratio is obtained for each animal. This step reduces inaccuracies since one value is generated based on proteins within the same sample. Apart from being independent of protein concentration, determining ratios has two additional advantages. Firstly, calculating ratios enlarges the differences between control and stressed animals, making interpretation easier. Secondly, ratios are based on the values of at least two biomarkers and thus enhance the reliability of the outcome.

#### 4.3.3. Effect of Contamination

Chewing, biting, and oral health problems can cause small wounds in pigs’ oral cavities, which can leak blood during saliva collection. Some candidate biomarkers, like alpha-2-HS-glycoprotein, cortisol, and α-amylase, are present in low concentrations in saliva but in much higher concentrations in blood [46,47]. Consequently, even minimal blood contamination can lead to artificially high levels of these components in saliva [48]. Additionally, hemoglobin can interfere with the determination of salivary testosterone and oxidative stress marker concentrations [49,50]. Some of the pitfalls of blood contamination may be clear. Nonetheless, considering these requires the amount of blood to be detectable in the salivary sample. Unfortunately, such a detection method is currently not available. Visual inspection of saliva samples for discoloration due to blood is probably the most frequently used method since a volume contamination of 0.1-0.2% blood already results in a tinted sample. Dipstick tests used to detect hemoglobin in urine have been suggested, too. Unluckily, false positive results often occur since the peroxidase in saliva also catalyzes the reaction on which the dipstick tests are based [51,52].

Both hemoglobin and albumin have been suggested as blood contamination markers. After all, the concentrations of these proteins are much higher in blood compared to saliva. Still, determining these proteins presents some sensitivity and reproducibility concerns, and several factors influence their concentrations [51]. Additionally, these proteins have been suggested as biomarkers for acute stress [52,53]. Moreover, they were present in higher concentrations in the salivary samples of chronically stressed piglets [35]. This stress-related upregulation hinders their possible use as biomarkers for blood contamination in the context of stress research. Serotransferrin is another protein that is present in higher concentrations in blood than in saliva. Although it is known that several factors, such as age, the levels of gonadal hormones, salivary flow rate, and chewing affect serotransferrin levels in saliva [54], it is recommended as the best indicator for blood contamination [51], complemented with visual inspection of the saliva sample.

Feed, dirt, or fecal material contamination can predominantly occur in farm settings. As these contaminants may increase the total amount of protein in the saliva sample, determining this parameter as a proportion of the sample’s protein concentration, thus irrespective of the saliva volume as with mass spectrometry, is problematic. In addition, feed particles within the oral cavity may interfere with the composition of the sampled saliva. For example, the concentration of α-amylase, a digestive enzyme responsible for starch cleavage, may be decreased by ingesting carbohydrate-rich feed. Indeed, adding commercial feed to porcine saliva samples reduces the salivary α-amylase concentrations, with higher levels of contamination resulting in a stronger reduction [41]. A study on equine saliva revealed that feed contaminants such as oats, grass, and hay affect the salivary components differently. In addition, several types of food contamination influence the biomarkers suggested for physical and psychosocial stress in pigs, such as α-amylase, adenosine deaminase, and total esterase [55]. Whether diverse feeds affect the salivary composition of porcine saliva differently should be further explored. In piglets, special attention could be paid to the effect of milk.

The effect of fecal contamination on the determination of several salivary components has only been studied to some extent. In pigs, cortisol concentrations increase in a dose-dependent matter. However, this only applies when the analysis is performed with an alphaLISA^®^ and not when a chemiluminescence assay is used. This once again draws attention to the variation that can be introduced by several analytical methods. Oxytocin, total esterase activity, and the total protein concentration also increase, α-amylase concentration drops, and the concentration of adenosine deaminase remains constant when a porcine salivary sample is contaminated with feces [41].

Additionally, discoloration of the sample may also impact spectrophotometric methods [41,55,56]. Some researchers report that sampling was postponed if animals ate or drank at the scheduled time since these acts might muddle the samples [38]. Oral rinsing before saliva collection therefore seems beneficial [55]. Alternatively, the sample can be purified via centrifugation, filtration, or chemical clarification with chitosan. Centrifugation appears to be the best option [41]. The downside of this technique is that it could remove macromolecular aggregates or proteins bound to bacteria or mucus [57].

It can be concluded that contamination can influence the results of saliva analyses in a variety of ways. Although this source of variation should be further explored, attempting to collect clean samples and visual inspection for tinted samples is paramount.

## 5. Salivary Biomarkers to Identify Stress

During the last two decades, more insight has been gained into the composition of porcine saliva. In addition, the effect that different conditions, such as acute and chronic physical and psychosocial stress, diseases, and inflammatory processes, may have on it has been elaborated. Moreover, many porcine salivary biomarkers for these conditions have been detected. However, as extensively pointed out in this review, many factors can influence the concentrations of these biomarkers. Therefore, more research is needed to identify the extent of these varying factors. Reference intervals should be set for every different combination of factors.

It should be noted that more validation is needed to deduce a set of salivary biomarkers that can detect stress in animals. Since no universal stress response exists, multiple aspects of the individual’s response to stress should be monitored. The advantage of saliva as a biological matrix is that it can indicate both the biological response to stress and the consequence of it [58]. As regards the biological response, the most often used biomarkers for stress are the products of direct activation of the SAM and HPA axis, i.e., α-amylase and chromogranin A on the one hand and cortisol on the other. Although their utility has drawbacks, these should be considered for the salivary biomarker panel to identify stress. These so-called activation biomarkers need to be complemented with consequence indicators. In this respect, odorant-binding protein (although also affected by disease), testosterone, and vomeromodulin-like protein should be further explored since a decrease in their concentration reflects a reduction in reproductive capacity. Besides this effect of stress, it has been proven that it also has immunosuppressive consequences, which can be detected in saliva using biomarkers. More specifically, reduced concentrations of immune-related markers could indicate chronic physical and psychosocial stress. Lower concentrations of chitinase, lipocalin-1 (although also affected by disease [40]), long palate lung and nasal epithelium protein 5, but also salivary lipocalin (although also affected by disease [40]), adenosine deaminase, protein S100-A9, and S100-A12 were found in lower concentrations in the saliva of stressed pigs [53]. However, lower immunity may result in higher infection rates and more diseases, leading to an altered immune response. Additionally, while disease is often a consequence of stress, it will also cause reduced welfare. Thus, biomarkers associated with infection and disease should be included in the assay. For example, most disease conditions correlate with elevated C-reactive protein [59]. Although its saliva concentrations are influenced by a circadian rhythm, age, sex, and breed, this biomarker could be used to indicate a diseased state [60,61]. It is noteworthy that adenosine deaminase, protein S100-A9, and S100-A12 are downregulated by acute stress but upregulated by disease and inflammation, suggesting that these proteins could have multiple purposes [40].

The advantage of saliva as a biological matrix is that it usually does not need further processing. If a collection technique does not require centrifugation, like the Micro•SAL™ (Oasis Diagnostics, Vancouver, WA, USA) salivary collection device, saliva can either be analyzed immediately or be stabilized for transport to clinical laboratories through preservatives. In the laboratory, analysis can be performed with ELISA, TR-IFMA, or enzymatic assays. However, finding and validating or developing assays that enable the quantification of recently discovered biomarkers is key. ELISA kits for porcine serum/plasma alpha-2-HS-glycoprotein, chitinase, and lipocalin-1 have been marketed but are not yet validated for porcine saliva. Unfortunately, assays for long palate lung and nasal epithelium protein 5, odorant-binding protein, and vomeromodulin-like protein are not commercially available.

As alternatives for lab-dependent analytical techniques, several options for on-site saliva analysis are possible. The most well-known portable sample analysis tool is the lateral flow test, which is usually used to detect the presence or absence of a specific biomarker or pathogen, like in a urine pregnancy (hCG) or a COVID-19 (coronavirus) test, respectively. Although the described tests are qualitative (positive or negative result), the lateral flow test has been modified now to generate a quantitative result. Usually, these quantitative tests require cartridges and specialized reading tools or a smartphone with a calibrated camera [62]. One of the downsides of this technique is that only one biomarker, for example, cortisol, can be quantified at a time.

Multiplexed immunoassays or lab-on-a-chip devices that detect several biomarkers at once could solve this problem. Such a multiplexing technique is combined with a portable point-of-service device capable of rapid, sensitive, automated, and multiple biomarker detection using human saliva [63,64].

## 6. Value of Salivary Stress Markers in Welfare Assessment

The value of salivary biomarkers to identify stress was discussed above. If the issues with variation and contamination can be overcome or when biomarkers that are less sensitive to variation are selected, saliva is the go-to matrix for physiological assessments in welfare research.

However, saliva should probably not be analyzed exclusively, although the potential of saliva as a biological matrix to assess porcine welfare is high. The welfare status of an animal is a complex condition that requires a broad panel of parameters to determine it. However, it is not always possible to determine a whole battery of parameters. Behavior assessment is an especially specialized discipline. Therefore, determining the presence of chronic stress could focus more on physiological assessments.

Since contemporary ethologists have abandoned the narrow-minded idea of monitoring welfare to identify and prevent suffering, oxytocin could be added to the salivary biomarker panel. A reduction of this salivary biomarker could indicate reduced welfare, whereas increased values could suggest the opposite, i.e., positive welfare [55]. Indeed, recent advances in ethology have resulted in a broader view of the welfare concept and have emphasized the importance of positive experiences. This relatively new concept of positive welfare is still not well defined, but adding positive welfare biomarkers to the saliva biomarker panel will certainly be of added value.

The cost of rapid, sensitive, automated, and multiple biomarker detection using porcine saliva may pose an issue, especially for welfare controls in the framework of obtaining welfare labels. Although 82% of the respondents of a European survey declared that the welfare of farm animals should be better protected and that higher transparency regarding housing and living conditions is needed, the willingness to pay for better housing conditions and welfare monitoring remains an obstacle. Surprisingly, willingness to pay for welfare-friendly animal-derived food products was the lowest for pigs, followed by fish, broilers, laying hens, dairy cows, and beef cows [65].

## 7. Conclusions

The research on welfare assessment tools for pigs knows a rich tradition, and yet no gold standard exists today. Many different indicators for chronic stress have their benefits but also their drawbacks. The present review demonstrates that analyzing the salivary proteome by means of proteomics is valuable for assessing the welfare status of pigs. Six potential salivary biomarkers for chronic stress are proposed. These biomarkers can be regarded as consequence indicators. In the case of stress, they are related to a reduced HPG axis response (odorant-binding protein and vomeromodulin-like protein), an immunosuppressive status (chitinase, lipocalin-1, and long palate lung and nasal epithelium protein), and a heightened anti-inflammatory response (alpha-2-HS-glycoprotein). Consequently, it is suggested that a panel of different biomarkers reflecting different affected pathways is valuable to detect stress. Moreover, if the ratios of the obtained values are considered, the strength of the salivary test increases. Unsurprisingly, further validation and more readily available analytical techniques should be developed.

## Figures and Tables

**Table 1 animals-14-01703-t001:** Overview of salivary biomarkers for physical and psychosocial stress in pigs and their influencing factors. If the box remains blank, to our knowledge, no information is available.

Biomarker	Up-/Down-Regulation	Acute/Chronic Stress	Other Conditions	Circadian Rhythm	Storage Information	Influence of Collection Device	Effect of Gland Distribution	Effect of Age, Sex, Breed, Season or Estrus Cycle
**Vomeromodulin-like protein**	Down	Chronic						Age
**Alpha-2-HS-glycoprotein**	Up	Chronic					No	Age
**Chitinase**	Down	Chronic	Asthma				No	Age
**Lipocalin-1**	Down	Both	Disease				No	Age, sex, and estrus cycle
**Long palate lung and nasal epithelium protein 5**	Down	Chronic						Age
**Odorant-binding protein**	Down	Both	Disease				No	Age and sex
Salivary lipocalin	Down	Acute					No	Sex
IgA	Up	Acute	Infection			Yes		No breed effect
IgM	Down	Acute	Infection					
α-amylase	Up	Acute		No	<4 days (4 °C), <3 months (−20 °C)	Yes	No	Age, no sex effect
IL-18	Up	Acute				Yes		
Chromogranin A	Up	Both		No	2 days (4 °C), 1 month (−20 °C), up to 7 freeze-thaw cycles			Season, no age (17 vs. 21 weeks) and sex effect
Serum amyloid A	Up	Both					No	
Testosterone	Up	Acute		No		Yes		
Albumin	Up	Acute	Infection and inflammation				No	
Cortisol	Up	Both	Physical activity	Yes	3 months (5 °C)	Yes		Age, sex, and breed
Prolactin inducible protein	Down	Acute					No	
Adenosine deaminase	Down	Both	Lameness, rectal prolapse, fatigue, inflammation (up)	Yes	4 days (4 °C), 1 month (−20 °C)			Age, sex, breed
Carbonic anhydrase IV	Up	Presumably both	Snaring (inconsistent), non-infectious growth rate retardation				Yes (higher concentrations in parotid saliva)	No age (2 vs. 4 weeks) effect, effect of estrus cycle
Protein S100-A8, calgranulin A, calprotectin	Up	Acute	Inflammation, immune-mediated diseases and sepsis (up)					
Protein S100-A9, calprotectin, calgranulin B	Down	Acute	Inflammation, immune-mediated diseases and sepsis (up)					
Protein S100-A12, calgranulin C	Down	Acute	Inflammation, immune-mediated diseases and sepsis (up)	Yes			Yes (higher concentrations in parotid saliva)	Age
Double headed protease inhibitor SMG	Up	Acute						
Haemoglobin	Up	Both	Lameness					
Total esterase activity	Up	Both	Pain discomfort		<1 day (4 °C), <1 month (−20 °C)			
Butyrylcholinesterase	Up	Both	Pain discomfort		<1 day (4 °C), <1 month (−20 °C)			
Lipase	Up	Both	Pain discomfort		<1 day (4 °C)			
Oxytocin	Down	Both						
Total protein concentration	Up	Acute		Yes			Yes	Effect of age, not of sex

## Data Availability

There are no new data created in the paper.

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
