# Peer review of "Welfare Assessment in Pigs Using the Salivary Proteome"

_animals, 2024, doi:10.3390/ani14111703_

Round 1

Reviewer 1 Report

Comments and Suggestions for Authors

This is a very thorough review paper on saliva and assessing welfare in pigs. Every possible avenue of a literature review is included and each of those parts are well written and comprehensive, however, I feel there is way more information in this manuscript than needed based on the title. I suggest condensing the material down to those sections that strictly have to do with whether or not saliva is a good welfare assessment in pigs. Just an example of a few sections possibly not needed; the saliva chemistry/anatomy, stress history, indicators of stress, other measurements of welfare - all of these are not needed to say is saliva is a good indicator of welfare. The manuscript should focus on  on studies that use saliva, which measurements, and if those results are accurate and valid, storage, etc. to make it a good assessment tool. T

Author Response

Dear reviewer,

Thank you for your constructive comments on our manuscript that we resubmit herewith. We have deleted entire chapters as well as individual sentences. In the latter case, the original sentences have been marked in red and crossed out. The manuscript is now much shorter and more focused on the proteomics part. Please find our answers to your detailed comments below in italics.

Reviewer 1

This is a very thorough review paper on saliva and assessing welfare in pigs. Every possible avenue of a literature review is included and each of those parts are well written and comprehensive, however, I feel there is way more information in this manuscript than needed based on the title. I suggest condensing the material down to those sections that strictly have to do with whether or not saliva is a good welfare assessment in pigs. Just an example of a few sections possibly not needed; the saliva chemistry/anatomy, stress history, indicators of stress, other measurements of welfare - all of these are not needed to say is saliva is a good indicator of welfare. The manuscript should focus on studies that use saliva, which measurements, and if those results are accurate and valid, storage, etc. to make it a good assessment tool. We have deleted the entire chapter on the definition of saliva and the paragraph on the history of stress. “6. Currently available methods to monitor stress” was also removed, as well as the original paragraphs “7.2.1. Sample collection” and “7.2.2. Samples treatment and storage”. From paragraph “7.2.5. Interpretation and comparison of salivary biomarker concentrations” we have only retained the bullet on the effect of contamination and incorporated it into the original paragraph 7.2.3. “3. Brief introduction to proteomics has been shortened substantially and is now the first paragraph of “The salivary proteome”.

Reviewer 2 Report

Comments and Suggestions for Authors

Este es un manuscrito que trata un tema muy interesante de importancia científica y social. However, some points should be addressed and corrections made, before this manuscript could be considered for publication.

1. The abstract should be more focused and reduced.

2. In the Introduction it would be recommended to mention and comment other reviews previously published about the use of saliva as a source of biomarkers of stress and welfare assessment in pigs such as:

*https://bmcvetres.biomedcentral.com/articles/10.1186/s12917-022-03176-w

*https://bmcvetres.biomedcentral.com/articles/10.1186/s12917-016-0791-8

The first review is only marginally commented and the second one seems that it is not included in the manuscript. These reviews could really complement the manuscript by supporting the interest of the field and should be more commented (even in the first review there are guidelines for stress biomarkers interpretation in saliva) and on the other hand that this current review will deal with the biomarkers of stress from a proteomic point of view. For example the second review could be of benefit for the point 6.2.

3. Lines 238 and 241 could be moved to introduction.

4.  At point 5 it should be clarified why it is indicated “the modern concept of stress”.

5. Faeces have various advantages also that should be commented in lines 709-719

6. To focus more the review in the proteomic field maybe the point 7.2 could be focused only in the points:

- 7.2.3. giving emphasis to proteomics but indicating that in addition other assays can be applied in field conditions and also include here all the importance that contamination can have in the proteomics or other analysis from the technical point on view

- 7.2.4. since it is related with the proteomic technique also

7. Points 7.3 and 7.4 could be put together since both deal on the same topic and their contents can be complemented.

Author Response

Dear reviewer,

Thank you for your constructive comments on our manuscript that we resubmit herewith. We have deleted entire chapters as well as individual sentences. In the latter case, the original sentences have been marked in red and crossed out. The manuscript is now much shorter and more focused on the proteomics part. Please find our answers to your detailed comments below in italics.

Reviewer 2

  1. The abstract should be more focused and reduced. The length of the abstract has been reduced substantially. The panel of six suggested salivary biomarkers of stress has been included. The simple summary has been reduced as well.

  1. In the Introduction it would be recommended to mention and comment other reviews previously published about the use of saliva as a source of biomarkers of stress and welfare assessment in pigs such as:

*https://bmcvetres.biomedcentral.com/articles/10.1186/s12917-022-03176-w

*https://bmcvetres.biomedcentral.com/articles/10.1186/s12917-016-0791-8

We have included and commented on the papers by Cerón et al., 2022 and Martínez-Miró et al., 2016 in the Introduction.

The first review is only marginally commented and the second one seems that it is not included in the manuscript. These reviews could really complement the manuscript by supporting the interest of the field and should be more commented (even in the first review there are guidelines for stress biomarkers interpretation in saliva) and on the other hand that this current review will deal with the biomarkers of stress from a proteomic point of view. For example the second review could be of benefit for the point 6.2. Chapter 6.2. no longer exists as it has been deleted upon request by reviewer 1.

  1. Lines 238 and 241 could be moved to introduction. We have moved these sentences to the end of the introduction.

  1. At point 5 it should be clarified why it is indicated “the modern concept of stress”. This comment is no longer valid as we have deleted the paragraph on the history of stress and thus do not need the title “The modern concept of stress” any longer to define the term stress. We have changed the title “5. The modern concept of stress” to “2. The definition of stress”.

  1. Faeces have various advantages also that should be commented in lines 709-719. Upon request by reviewer 1, chapter “6. Currently available methods to monitor stress” was removed, which means that also the discussion on feces as a stress monitoring tool has been deleted.

  1. To focus more the review in the proteomic field maybe the point 7.2 could be focused only in the points:

- 7.2.3. giving emphasis to proteomics but indicating that in addition other assays can be applied in field conditions and also include here all the importance that contamination can have in the proteomics or other analysis from the technical point on view

- 7.2.4. since it is related with the proteomic technique also

We have deleted the paragraphs “7.2.1. Sample collection” and “7.2.2. Samples treatment and storage”. From paragraph “7.2.5. Interpretation and comparison of salivary biomarker concentrations” we have only retained the bullet on the effect of contamination and incorporated it into the original paragraph 7.2.3.

  1. Points 7.3 and 7.4 could be put together since both deal on the same topic and their contents can be complemented. 7.3 and 7.4 have been put together in what is now “4. Salivary biomarkers to identify stress”. The last chapter is now “5. Value of salivary stress markers in welfare assessment” and has been modified substantially.

Reviewer 3 Report

Comments and Suggestions for Authors

The authors reviewed the biological matrices that can be collected to evaluate the response to stress conditions in piglets and estimate the level of animal well-being. They showed the advantages and disadvantages of each matrix. A huge part of the review was focused on saliva and all the compounds (steroids, proteins, etc.) that can be measured to establish the level of animal well-being. However, many factors can influence the assessment of most of these factors. Finally, the review proposed six potential salivary biomarkers for chronic stress. The review is well written however many paragraphs are unnecessary because the topics have already been reviewed in manuscripts focused on specific arguments. Thus I suggest removing these paragraphs, citing only the data necessary for the present manuscript's aim and referring to specific reviews for a deep investigation of the topic. I also suggest revising the order of presentation of the different arguments to increase the readability of the manuscript.  It could be much better to move the 5th chapter (definition of stress and response to stress stimuli) immediately after the introduction that has to be modified accordingly to the new order of exposition. Then the authors can continue with the methods to monitor stress levels showing that among the other saliva sampling could be the most useful matrix to measure animal well-being in the pigs. Thus you can define and explain saliva composition and all the different glands that contribute to its secretion (i.e. the current chapter 2). Following you can insert chapter 3 that can be reduced (there are some very technical paragraphs that reduce the readability for not expert readers and are not relevant for the aim of the review). I think that 4th chapter can be deleted, inserting the last sentence in the chapter of stress definition. Chapter 7 will remain the last one.

Some abbreviations were not explained (e.g. H&E staining; TR-IFMA)

Minor points:

lines 92-93 what did the authors want to explain with this sentence? Utilizing specific sequences you can easily amplify and evaluate only specific genes or DNA regions, thus the presence of bacteria is not so problematic. Please explain.

Line 204 Did you want to write "unstimulated"?

lines 261-265 can be synthesised in one sentence avoiding repetition 

Table1 The authors can bold the protein suggested to be the most useful salivary biomarkers for chronic stress. 

Author Response

Dear reviewer,

Thank you for your constructive comments on our manuscript that we resubmit herewith. We have deleted entire chapters as well as individual sentences. In the latter case, the original sentences have been marked in red and crossed out. The manuscript is now much shorter and more focused on the proteomics part. Please find our answers to your detailed comments below in italics.

Reviewer 3

  • Many paragraphs are unnecessary because the topics have already been reviewed in manuscripts focused on specific arguments. Thus I suggest removing these paragraphs, citing only the data necessary for the present manuscript's aim … We have removed a number of unnecessary chapters as suggested by reviewer 1. The deleted chapters include “2. The definition of saliva”, “4. The conceptualization of stress through history”, “6. Currently available methods to monitor stress”. The original paragraphs “7.2.1. Sample collection” and “7.2.2. Samples treatment and storage”. From paragraph “7.2.5. Interpretation and comparison of salivary biomarker concentrations” we have only retained the bullet on the effect of contamination and incorporated it into the original paragraph 7.2.3. Chapter “3. Brief introduction to proteomics” was massively condensed and is now the first paragraph of “The salivary proteome as a tool to monitor stress”.

  • … and referring to specific reviews for a deep investigation of the topic. We have included two review papers, i.e., Cerón et al., 2022 and Martínez-Miró et al., 2016, on the use of saliva as a source of biomarkers of stress and welfare assessment in pigs in the Introduction.

  • I also suggest revising the order of presentation of the different arguments to increase the readability of the manuscript. It could be much better to move the 5th chapter (definition of stress and response to stress stimuli) immediately after the introduction … We have moved this 5th chapter that now is the 2nd

  • … that has to be modified accordingly to the new order of exposition. We have rearranged several chapters and the introduction is modified accordingly.

  • Then the authors can continue with the methods to monitor stress levels showing that among the other saliva sampling could be the most useful matrix to measure animal well-being in the pigs. Thus you can define and explain saliva composition and all the different glands that contribute to its secretion (i.e. the current chapter 2). We have deleted the original chapters 2 and 6 on the definition of saliva and on the methods to monitor stress and welfare, as suggested by reviewer 1.

  • Following you can insert chapter 3 that can be reduced (there are some very technical paragraphs that reduce the readability for not expert readers and are not relevant for the aim of the review). The original chapter 3 on the proteomics has been reduced substantially.

  • I think that 4th chapter can be deleted, inserting the last sentence in the chapter of stress definition. We have deleted the original 4th chapter and incorporated the last sentence in the paragraph entitled “Introduction” within the section “The definition of stress”.

  • Chapter 7 will remain the last one. The last chapter is now “5. Value of salivary stress markers in welfare assessment” and has been modified substantially.

  • Some abbreviations were not explained (e.g. H&E staining; TR-IFMA) H&E staining is no longer mentioned since we have removed the chapter in which it was present. We have put the full term for which TR-IFMA stands in between brackets, i.e., (Time-resolved immunofluorometric assay), the first time it is mentioned in the manuscript.

  • lines 92-93 what did the authors want to explain with this sentence? Utilizing specific sequences you can easily amplify and evaluate only specific genes or DNA regions, thus the presence of bacteria is not so problematic. Please explain. This comment is no longer valid since we have deleted chapter “2. The definition of stress”.

  • Line 204 Did you want to write "unstimulated"? This comment is no longer valid since we have deleted chapter “2. The definition of stress”.

  • lines 261-265 can be synthesised in one sentence avoiding repetition. The entire chapter on proteomics have been condensed.

  • Table1 The authors can bold the protein suggested to be the most useful salivary biomarkers for chronic stress. Thank you for the suggestion. We have put the most useful proteins in the first six rows of the table and put them in bold.

Round 2

Reviewer 1 Report

Comments and Suggestions for Authors

Thank you for your improved revision better focusing on the relevant information for a salivary - welfare assessment review.

Author Response

Dear reviewer,

Thank you for your appreciation.

Reviewer 3 Report

Comments and Suggestions for Authors

The authors have improved the manuscript however I still believe that the order of presentation could be further enhanced by moving chapter 5 at the end of chapter 2. However, lines 123-125 should be moved at the end of the review and inserted in the conclusion paragraph. 

Author Response

Dear reviewer,

Thank you for your final thoughts. Below you can find our answers to your comments.

  • The authors have improved the manuscript however I still believe that the order of presentation could be further enhanced by moving chapter 5 at the end of chapter 2. We have not moved the entire chapter 5 to the end of chapter 2 because we think that some information in the original chapter 5 can only be provided at the end of the manuscript. For example the inclusion of oxytocin that is a biomarker of positive welfare to the panel of salivary biomarkers. In addition, we like to keep the original chapter 5 as it builds on the previous chapter that discusses the biomarkers for stress. In the original chapter 5, we broaden the discussion by not considering just stress but welfare, and the place salivary protein biomarkers may have in a welfare assessment protocol. However, the information that could be moved to the end of chapter 2 has now been included in a short additional chapter (so the chapter numbering has been adapted accordingly) on the value of saliva as a biological matrix to assess welfare. 
  • However, lines 123-125 should be moved at the end of the review and inserted in the conclusion paragraph. We have moved these lines and reorganized some sentences in the discussion in order to remain a good flow in the text.

All moved text is highlighted in yellow, additional text is stressed in green and deleted text is crossed in red.

Yours sincerely,

Christophe Casteleyn